# Scientific Algorithm Discovery by Augmenting AlphaEvolve with Deep Research

## Abstract

Large language models hold promise as scientific assistants, yet existing agents either rely solely on algorithm evolution or on deep research in isolation, both of which face critical limitations. Pure algorithm evolution, as in AlphaEvolve, depends only on the internal knowledge of LLMs and quickly plateaus in complex domains, while pure deep research proposes ideas without validation, resulting in unrealistic or unimplementable solutions. We present DeepEvolve, an agent that integrates deep research with algorithm evolution, uniting external knowledge retrieval, cross-file code editing, and systematic debugging under a feedback-driven iterative loop. Each iteration not only proposes new hypotheses but also refines, implements, and tests them, avoiding both shallow improvements and unproductive over-refinements. Across nine benchmarks in chemistry, mathematics, biology, materials, and patents, DeepEvolve consistently improves the initial algorithm, producing executable new algorithms with sustained gains. By bridging the gap between unguided evolution and research without grounding, DeepEvolve provides a reliable framework for advancing scientific algorithm discovery.

## 1 Introduction

Large language models (LLMs) are emerging as foundation models for building AI scientists, automating processes such as lab work, mathematical discovery, and ML research (Boiko et al., 2023; Chan et al., 2024). Many scientific problems are difficult to solve but easy to evaluate (Romera-Paredes et al., 2024), raising hope that LLMs can drive algorithm discovery through reasoning, planning, and execution. Recent progress shows advances in ML benchmarks (Chan et al., 2024), mathematical discovery (Novikov et al., 2025), and experimental design (Boiko et al., 2023). However, it is still challenging for LLM-based agents to push algorithmic frontiers by not only generating new hypotheses (Gottweis et al., 2025) but also implementing them as working code.

The combination of hypothesis generation with code execution and evaluation has been explored in systems such as FunSearch (Romera-Paredes et al., 2024) and AlphaEvolve (Novikov et al., 2025), with the latter achieving breakthroughs in $4 \times 4$ matrix multiplication. AlphaEvolve uses an ensemble of LLMs to generate code that encodes new scientific hypotheses. However, its generalization to broader domains such as chemistry, biology, and materials remains uncertain. These domains present vast, unbounded search spaces, where relying solely on LLMs themselves is unlikely to yield substantive algorithmic advances. A preliminary study of molecular property prediction is shown at the top of Figure 1. Pure algorithm evolution with AlphaEvolve[1] yields limited improvement $(0.791 \rightarrow 0.797)$, only 0.6% after 100 iterations. Surprisingly, the best algorithm appears in the first generation evolved from the initial algorithm, outperforming the other 24 candidates with deeper generations. Some deeply evolved algorithms, including the second-best one, show only marginal improvements after multiple refinements of the initial idea.

From the figure, we find that high-quality idea generation can be a bottleneck for algorithm evolution in broader scientific domains. To address this, we augment the evolution system with deep research, a framework designed for intensive knowledge work that requires thorough and reliable retrieval from the internet. General deep research methods (Xu & Peng, 2025) synthesize information from diverse online sources for scientific hypothesis generation but lack feedback from hypothesis testing. This

---

[1]The code of AlphaEvolve is unavailable; we follow an open-source reproduction (Sharma, 2025).

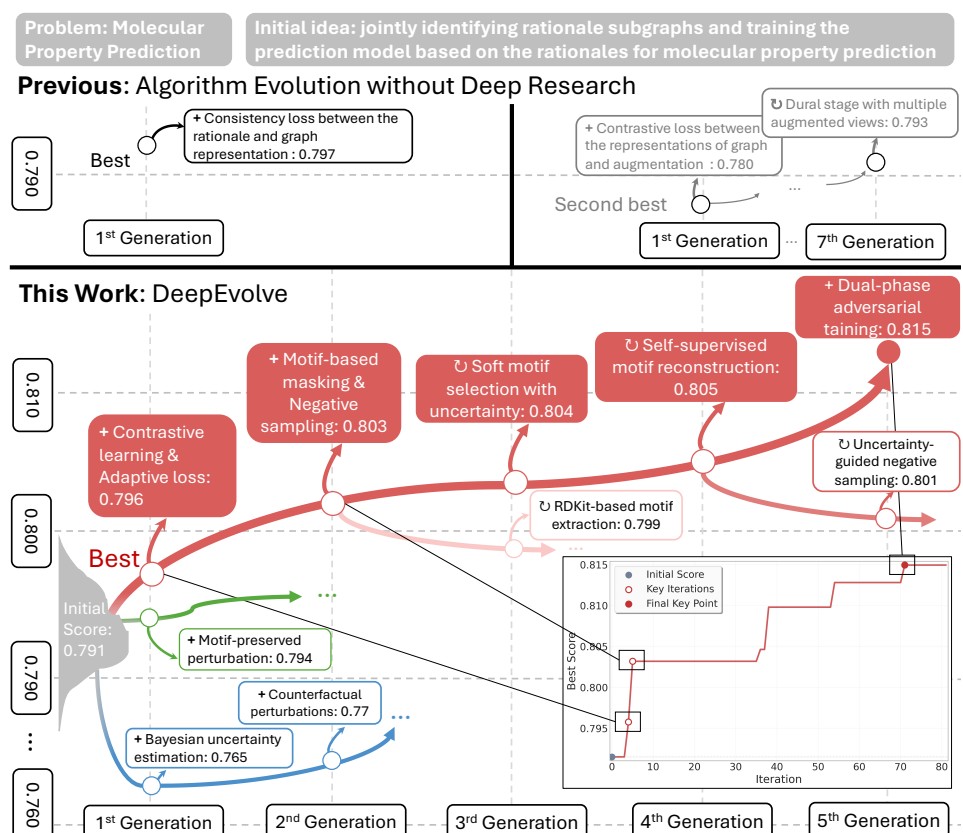

Figure 1: The top panel shows AlphaEvolve-style pure algorithm evolution without deep research, where the best improvement appears in the first generation and later iterations have marginal gains. The bottom panel shows DeepEvolve, which integrates deep research. DeepEvolve avoids shallow or excessively deep but unproductive evolutions, achieving sustained progress with clear performance jumps at key iterations. $+$ denotes adding a new idea, and $\circlearrowleft$ denotes refining a previous idea.

may lead to proposals that are too difficult or unrealistic to implement. To address this limitation, we perform deep research on a specific algorithm, accompanied by inspiring algorithms that have been successfully implemented in past discoveries. We instruct deep research to generate research proposals with pseudo-code that are easy to implement in the early stages, while moving toward higher-impact ideas in later stages. Proposals for an algorithm often involve modifying multiple code files, such as those for data preprocessing or model architecture. This requires the coding agent to parse and analyze across files, a capability added to our design but absent in AlphaEvolve, which substantially increases coding difficulty. A debugging agent is thus introduced to resolve errors during execution, further improving the success rate of algorithmic implementation (Table 3). Finally, the evaluation function tests the algorithm proposal and provides feedback to deep research for the next proposal. As shown at the bottom of Figure 1, this approach produces clear improvements over both the initial algorithm and pure algorithmic evolution. Unlike shallow evolutions or overly deep but marginal ones, deep research balances depth and yields clear performance jumps at key iterations.

In this work, we propose DeepEvolve to orchestrate algorithmic deep research, implementation, evaluation, and evolution. The workflow, shown in Figure 2, has six components. The first three generate a research proposal by planning research questions, searching for answers online, and composing a proposal. This is then used as a prompt for the coding agent, which performs cross-file edits and multiple rounds of debugging. Each algorithm is evaluated and stored in a database that serves as long-term memory, providing candidates and inspiration for the next round of evolution.

We benchmark nine scientific problems across chemistry, mathematics, biology, materials, and patent domains, covering diverse data modalities such as molecules, geometries, partial differential

equations, and images (Table 1). Results show consistent improvements over existing algorithms, generating original and promising new methods (Figure 3) with high performance scores (Table 2).

## 2 Problem Definition for Algorithm Discovery

Let $P = (D, g)$ denote a scientific problem in domains such as mathematics, chemistry, or biology. Each problem has evaluation data $D = \{(q_i, a_i)\}_{i=1}^N$, where $q_i$ are questions and $a_i$ are ground-truth answers, and an evaluation function $g$ that compares the ground-truth answers with predicted answers. The score is computed as $s = g(\{a_i\}_{i=1}^N, \{\hat{a}_i\}_{i=1}^N)$. Here $\hat{a}_i$ are the outputs of an algorithm $f : Q \to A$ that maps each question $q_i$ to an answer $\hat{a}_i = f(q_i)$. Both computation and evaluation should be completed within bounded time (e.g., minutes or hours). We define a textualization function $\tau$ that converts structured objects into text. For example, $\tau(P)$ is the problem description as $\tau_P$ and $\tau(f)$ is the algorithm description as $\tau_h$. The goal of algorithm discovery is to optimize $f$ for higher $s$.

A problem instance in mathematics and geometry is the circle packing. The evaluation is to maximize the sum of radii for $n$ circles placed within a unit square. This can be formalized as a constrained problem $P$. The algorithm $f$ is a Sequential Least Squares Programming (SLSQP) solver, as shown in an open-source reproduction of AlphaEvolve (Novikov et al., 2025; Sharma, 2025). Different evaluation data correspond to different values of $n$, such as $n = 26, 27, \dots$.

A second example is molecular property prediction. The goal is to develop ML algorithms that train models to generalize well. They should also yield interpretable predictions for each molecule. We study automated discovery of such algorithms across domains using research and coding agents.

## 3 DeepEvolve for Algorithm Discovery

DeepEvolve takes as input three things: a problem $P$, an initial algorithm $f$, and user instructions $u$. From these, DeepEvolve produces an updated algorithm. For a fixed problem and user instruction, we can think of an update operator that takes the current algorithm and returns a new one. This operator is built from six modules, applied in sequence: plan, search, write, code, evaluation, and evolutionary selection. Together, they transform the algorithm in a systematic way. The algorithm evolves by repeatedly applying this update operator. Starting with the initial version $f^{(0)} = f$, each new version is produced from the previous one. After $K$ rounds, we obtain a final candidate $f^{(K)}$. The best algorithm is chosen from all the intermediate versions $\{f^{(0)}, f^{(1)}, \dots, f^{(K)}\}$ by selecting the one that achieves the highest evaluation score on the given problem. In the following subsection, we first describe how the input context is built Section 3.1. We then introduce each component in Section 3.2 corresponding to Figure 2, detailing the synergy between deep research and algorithm evolution.

### 3.1 Input of Problem, Algorithms, and Instructions

**Problem as Input.** The input context of problem $P = (g, \mathcal{D}, \tau_P)$ includes three parts: the evaluation function $g$ implemented as code, the evaluation data $\mathcal{D}$, and a textual problem description $\tau_P$. Evaluation metrics associated with $g$ are summarized in Table 1. Given $g$ and $\mathcal{D}$, the optimization direction of the algorithm can be specified. The problem description $\tau_P$ consists of one or more paragraphs that define the task, relevant terminology, notations, equations, and evaluation metrics.

**User Instructions.** The user instructions $u$ contain a textual specification of user-defined requirements, providing additional guidance for algorithm evolution. While the evaluation metrics $g$ and data $\mathcal{D}$ determine the primary optimization objective, users may express auxiliary preferences or constraints such as desired research directions (e.g., efficiency, interpretability, generalizability), available software dependencies, hardware constraints, and runtime budgets.

**Algorithm as Input.** The algorithm $f$ consists of both the code implementation and its textual description $\tau_h$. Compared to AlphaEvolve (Novikov et al., 2025), we consider the algorithm implementation spanning multiple files with an entry point that computes the outputs for evaluation. Each algorithm description $\tau_h$ includes the motivation, a summary, pseudo-code, the performance $s$, and qualitative assessments such as originality, future potential, and implementation difficulty.

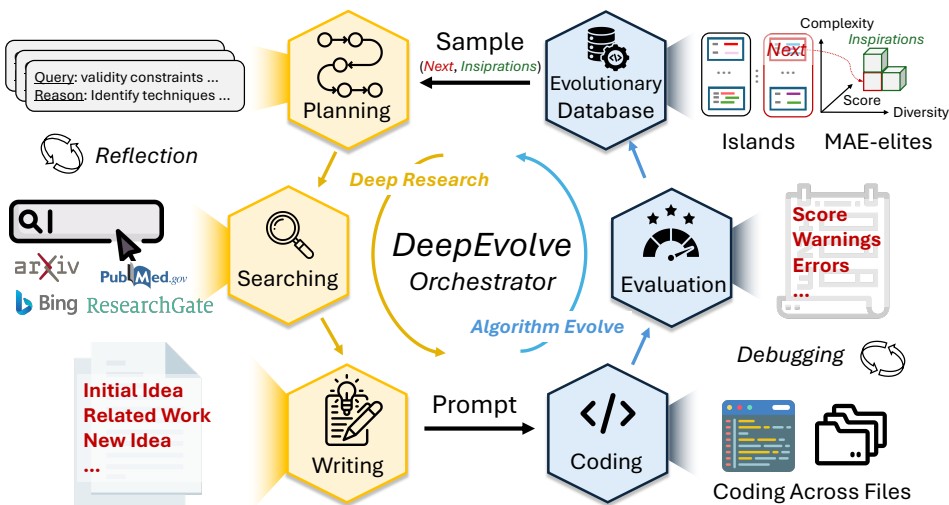

Figure 2: DeepEvolve is structured around six collaborative modules that alternate between deep research and algorithm evolution. Deep research generates informed hypotheses through planning, retrieval, and synthesis, while algorithm evolution translates these hypotheses into code, evaluates them, and applies evolutionary strategies for selection.

## 3.2 FRAMEWORK DESIGNS

In an iteration from $t$ to $t + 1$, we start from a candidate algorithm $f$ together with a set of inspiring algorithms $\{f_1^{\text{insp}}, f_2^{\text{insp}}, \ldots, f_n^{\text{insp}}\}$ and their evaluations to conduct deep research. This differs from a direct implementation (Xu & Peng, 2025), which brainstorms ideas without feedback. After proposing a new algorithm, it is implemented with functions distributed across multiple files and supported by automatic debugging. In contrast, AlphaEvolve (Novikov et al., 2025) designs algorithms directly with LLMs, evolves code within a single file, and lacks a code correction mechanism.

**Algorithmic Deep Research.** The planning step generates a small set of research questions that guide the direction of the next improvement. The agent is instructed to be more exploratory if the algorithm has already undergone multiple updates. These questions are then searched on websites, including sources such as PubMed and arXiv, and the results are summarized in a few paragraphs. Finally, a writing agent proposes a new algorithm by integrating the retrieved evidence with the input context (i.e., problem, algorithm, and inspirations). It is instructed to compare different methods and identify promising directions. A group of new ideas is generated with self-evaluation, and the most promising one is chosen as the final proposal based on the current evolutionary progress. In early stages, it prioritizes feasible ideas, while in later generations it emphasizes higher-impact ideas. Finally, it writes a short proposal for the new algorithm, including pseudo-code to guide the implementation.

**Algorithmic Implementation.** We use a coding agent to implement the proposed algorithm. It parses multi-file codebases using delimiters. It then localizes the minimal set of code regions that require modification and applies targeted updates to implement the proposed algorithm. However, it is easy for new code to contain bugs, especially when modifying different files such as those for data preprocessing and model architecture. During execution, error and warning messages provide valuable information for debugging. Therefore, we introduce a debugging agent to handle failures based on program execution feedback. Given a budget (e.g., five attempts), if execution remains unsuccessful after debugging, the algorithm is assigned a score of zero.

**Evaluation and Evolutionary Database.** The algorithm is scored ($s > 0$) once it is successfully executed and evaluated. We add it with the score to a database, which is maintained with evolutionary methods for sampling the next candidate and inspiring algorithms. We use island-based populations (Tanese, 1989) as the candidate pool for the next iteration. At each step, we sample an island and then select $f$ from it, favoring high-score candidates while retaining exploration. For inspirations, MAP-Elites (Mouret & Clune, 2015) samples nearby algorithms of $f$ based on three

Table 1: Benchmark tasks, data types, domains, and evaluation metrics. New scores are used for evaluation such that higher values indicate better performance.

| Problem | Description | Data Type | Domain | Original Metric | New Score | Source |
|---|---|---|---|---|---|---|
| Molecular Prediction | Molecular property prediction | Small molecule | Chemistry | AUC over multiple model initializations | $0.5 \cdot \text{AUC}_{\text{mean}} + 0.5 \cdot \text{AUC}_{\text{std}}$ | OGB (Hu et al., 2020) |
| Molecular Translation | Image-to-text translation of chemical structures | Image–molecule pair | Chemistry | Levenshtein distance | $1 - \text{Levenshtein distance}$ | Kaggle (Howard et al., 2021) |
| Circle Packing | Packing circles inside a unit square to maximize sum of radii | Geometry | Mathematics | Mean sum of radii with 26 to 32 circles | Same as Original | AlphaEvolve & Erich's Packing Center (Novikov et al., 2025) |
| Burgers' Equation | Solving Burgers' equation | Partial Differential Equation | Mathematics | Normalized RMSE (nRMSE) | $\frac{1}{\text{nRMSE} \cdot 10^3}$ | CodePDE (Li et al., 2025) |
| Parkinson's Disease | Disease progression prediction | Time series | Biology | Symmetric Mean Absolute Percentage Error (SMAPE) | Same as Original | Kaggle (Kirsch et al., 2023) |
| Nuclei Image | Nuclei segmentation from images | Image | Biology | Mean average precision (mAP) | Same as Original | Kaggle (Goodman et al., 2018) |
| Open Vaccine | mRNA vaccine degradation prediction | mRNA sequence | Biology | Mean column-wise RMSE (MCRMSE) | $\frac{1}{1+\text{MCRMSE}}$ | Kaggle (Das et al., 2020) |
| Polymer Prediction | Prediction of polymer properties | Polymer | Materials | Weighted MAE (wMAE) and $R^2$ | $\frac{1}{1+\text{wMAE}} \cdot 0.5 + R^2 \cdot 0.5$ | Kaggle (Liu et al., 2025) |
| USP P2P | Phrase-level semantic matching in patents | Text | Patent | Pearson correlation | Same as Original | Kaggle (Cenkci et al., 2022) |

features: performance score, code diversity, and code complexity. These features are mapped to cells in a grid, and neighboring cells are used as inspiration for future candidates $f$.

**Reflection** The reflection mechanism is applied in both algorithmic deep research and implementation as a quick checkpoint for potential issues. For deep research, a reflection agent decides whether to continue planning, continue searching, or update the writing report, subject to a maximum number of reflections. For coding, the agent performs self-reflection to check whether its code aligns with the proposed algorithm and to detect potential syntax errors.

In DeepEvolve, algorithmic deep research, implementation, and evaluation are coupled across multiple iterations. Deep research alone provides knowledge but no tested progress, while implementation and iteration alone explore ideas blindly without grounding in recent research. By linking the two, the process mirrors human discovery: informed by existing knowledge, tested through implementation, refined with feedback, and improved through repeated cycles. To integrate the iterations more compactly, we instruct the deep research agents based on evolutionary progress (early or mature) and algorithmic history with evaluation feedback. We also use multiple checkpoints (e.g., code modification, self-reflection, debugging) for the coding agent to verify whether its implementation aligns with the proposed algorithm. Empirically, we study how deep research, implementation, and evaluation reinforce each other through evolutionary optimization in Section 4.3.

# 4 EXPERIMENTS

We investigate three research questions (RQs): RQ1: Can DeepEvolve discover new algorithms that improve both effectiveness and efficiency across diverse tasks? RQ2: How do the deep research and coding agents interact during the discovery process? RQ3: We conduct ablations and case studies to examine the designs and performance of DeepEvolve.

## 4.1 SET-UPS

We include nine research problems spanning chemistry, mathematics, biology, and materials as summarized in Table 1. These problems involve diverse data modalities, including molecules, images, mRNA, text, time series, geometric structures, and multi-modal inputs. For consistent evaluation, we standardize evaluation metrics (e.g., AUC-ROC, RMSE, precision, Pearson correlation) defined in each problem into a common form as the new scores, where higher values indicate better performance.

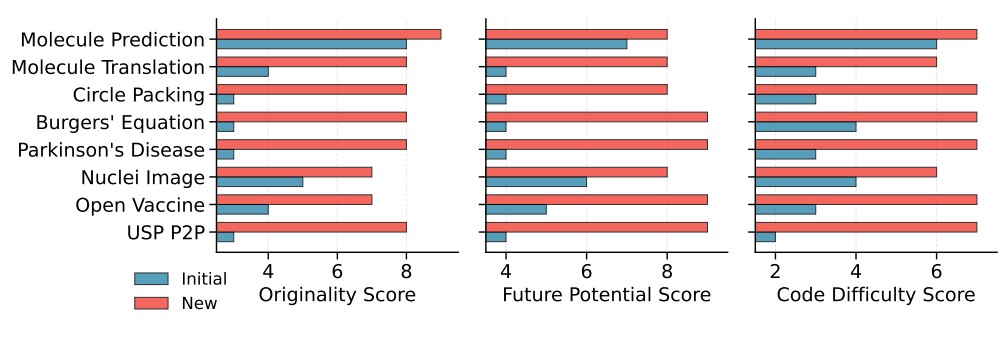

Figure 3: Evaluation of the idea from initial and new algorithms with LLM-as-a-judge.

Table 2: Quantitative comparison of new algorithms discovered by DeepEvolve with the initial ones in terms of effectiveness (new scores; see Table 1) and efficiency (runtime in minutes). Efficiency is not the primary optimization objective in DeepEvolve; it could be included in the user query.

| Problem | Performance with New Scores (↑) | | | Runtime in Minutes | | |
|---|---|---|---|---|---|---|
| | Initial Algorithm | New Algorithm | Improvement (%) | Initial Algorithm | New Algorithm | Reduced Time (Minutes) |
| Molecular Prediction | 0.7915 | 0.8149 | 2.96 | 5.06 | 7.64 | -2.58 |
| Molecular Translation | 0.1885 | 0.2562 | 35.94 | 21.42 | 5.44 | 15.98 |
| Circle Packing | 0.3891 | 2.9806 | 666.02 | 1.46 | 3.54 | -2.08 |
| Burgers' Equation | 0.6638 | 0.6666 | 0.42 | 12.77 | 23.35 | -10.58 |
| Parkinson's Disease | 0.5317 | 0.5876 | 11.82 | 1.26 | 22.05 | -20.79 |
| Nuclei Image | 0.3185 | 0.3405 | 6.91 | 11.37 | 10.61 | 0.76 |
| Open Vaccine | 0.7187 | 0.7214 | 0.39 | 26.68 | 14.40 | 12.28 |
| Polymer Prediction | 0.6770 | 0.7714 | 13.94 | 9.37 | 5.75 | 3.62 |
| USP P2P | 0.8036 | 0.8146 | 1.36 | 14.36 | 5.85 | 8.51 |

For each problem, we designate an initial algorithm as the baseline and apply DeepEvolve to optimize and generate new algorithms. For the molecule and polymer tasks, we improve the graph rationalization method GREA (Liu et al., 2022) in different directions specific to each problem. For the circle packing problem, we adapt the SLSQP algorithm from OpenEvolve (Sharma, 2025), an open-source implementation of AlphaEvolve. For the Burgers equation, we use the baseline provided by CodePDE (Li et al., 2025). For problems derived from Kaggle competitions, including molecular translation, Parkinson's disease progression, nuclei image segmentation, Open Vaccine, and USP P2P, we use baseline solutions provided by competition participants. More details are in appendix B.

To discover new algorithms, we define the primary optimization objective as the new scores in Table 1, with efficiency specified as a secondary objective in the prompt. The algorithm development process is constrained to a 30-minute time budget and a single GPU (2080-Ti or A6k). We evaluate both baseline and generated algorithms using quantitative metrics and qualitative analysis.

## 4.2 RQ1: Effectiveness and Efficiency for the Newly Discovered Algorithms

We conduct a quantitative analysis of how DeepEvolve improves the initial algorithms in terms of both effectiveness and efficiency. As shown in Table 2, DeepEvolve achieves improvements in both aspects on six of the nine tasks. In the remaining three cases, DeepEvolve generates algorithms that improve the primary performance objective while satisfying the 30-minute runtime constraint.

**The performance improvement achieved by DeepEvolve varies from 0.39% to 666.02%, depending on the problem type and the maturity of the initial algorithm.** In Circle Packing, the initial algorithm is designed for a fixed configuration (i.e., packing 26 circles) (Sharma, 2025) and fails to generalize to variable-sized constructions, often producing invalid solutions. In contrast, DeepEvolve discovers a new algorithm that generalizes across a broader range of circle counts while maintaining valid packings, resulting in a substantial performance gain. In other tasks, the improvement is relatively marginal due to different factors. The baseline for Burgers' Equation is based on a very recent state-of-the-art method (Li et al., 2025), leaving limited room for further improvement. For

```
 1  -  def forward(self, batched_data):
 1  +  ### >>> DEEPEVOLVE-BLOCK-START: Incorporate dual augmented views with contrastive loss and adaptive weighting in ACGR
 2  +  ### >>> DEEPEVOLVE-BLOCK-START: Incorporate dual augmented views with motif-aware attribute masking and adaptive weighting in ACGR
 3  +  ### >>> DEEPEVOLVE-BLOCK-START: Integrate self-supervised motif reconstruction branch with uncertainty-guided negative sampling
 4  +  ### >>> DEEPEVOLVE-BLOCK-START: Incorporate dual-phase adversarial perturbation and uncertainty-guided negative sampling in forward pass
 5  +  def forward(self, batched_data, phase="standard"):
 2  6      h_node = self.graph_encoder(batched_data)
 3  -      h_r, h_env, r_node_num, env_node_num = self.separator(batched_data, h_node)
 4  -      h_rep = (h_r.unsqueeze(1) + h_env.unsqueeze(0)).view(-1, self.emb_dim)
 5  -      pred_rem = self.predictor(h_r)
 6  -      pred_rep = self.predictor(h_rep)
 7  +      # Self-supervised motif reconstruction branch: apply motif-aware attribute masking
 8  +      masked_data = self.motif_mask(batched_data)
 9  +      h_masked = self.graph_encoder(masked_data)
10  +      # Reconstruction: recover masked motifs from the masked view
11  +      motif_pred = self.motif_decoder(h_masked)
12  +      loss_recon = 1 - F.cosine_similarity(motif_pred, h_node, dim=1).mean()
13  +
14  +      # If in adversarial phase, apply dual-phase perturbation based on computed uncertainty
15  +      if phase == "adversarial" and hasattr(self, "last_uncertainty"):
16  +          perturb = torch.randn_like(h_node) * (
17  +              self.last_uncertainty.mean() * self.gumbel_tau
18  +          )
19  +          h_node = h_node + perturb
20  +
21  +      # Generate dual augmented views via separator for environment replacement
22  +      h_r1, h_env1, r_node_num1, env_node_num1 = self.separator(batched_data, h_node)
23  +      h_r2, h_env2, r_node_num2, env_node_num2 = self.separator(batched_data, h_node)
24  +      pred_rem = self.predictor(h_r1)
25  +
26  +      # Compute contrast losses with uncertainty-guided negative sampling in adversarial phase
27  +      if phase == "adversarial":
28  +          adv_negatives = h_r1[torch.randperm(h_r1.size(0))]
29  +          contrast_loss_env = info_nce_loss(
30  +              h_r1, h_r2, temperature=self.temperature, negatives=adv_negatives
31  +          )
32  +      else:
33  +          contrast_loss_env = info_nce_loss(h_r1, h_r2, temperature=self.temperature)
34  +      contrast_loss_motif = info_nce_loss(
35  +          h_node, h_masked, temperature=self.temperature
36  +      )
37  +      contrast_loss = (contrast_loss_env + contrast_loss_motif) / 2
38  +
39  +      # Regularization to align node count ratios with the predefined gamma
40  +      r_node_num = (r_node_num1 + r_node_num2) / 2
41  +      env_node_num = (env_node_num1 + env_node_num2) / 2
 7  42      loss_reg = torch.abs(
 8  -          r_node_num / (r_node_num + env_node_num)
 9  -          - self.gamma * torch.ones_like(r_node_num)
43  +          r_node_num / (r_node_num + env_node_num) - self.gamma
10  44      ).mean()
11  -      output = {"pred_rep": pred_rep, "pred_rem": pred_rem, "loss_reg": loss_reg}
12  -      return output
45  +
46  +      output = {"pred_rem": pred_rem, "contrast_loss": contrast_loss, "loss_reg": loss_reg, "motif_loss": loss_recon}
47  +      return output
48  +  ### <<< DEEPEVOLVE-BLOCK-END
13  49      ...
```

Figure 4: The new `model.forward()` for Molecular Prediction. DeepEvolve proposes contrastive learning in Line 29-34, motif-aware masking in Line 8, and additional modules (see Figure 1) to improve the algorithm. The code of these functions is in appendix C.3.

Open Vaccine, model training requires more time and GPU resources, and we observe that evolving algorithms frequently exceed the 30-minute runtime budget, constraining DeepEvolve's search space.

**DeepEvolve improves algorithm originality and future potential, while the more complex implementation is handled through automatic code debugging.** We evaluate the quality of algorithmic ideas using an LLM-as-a-judge approach, assessing each from three dimensions: originality, future potential, and implementation difficulty. Language models (o3-mini) perform deep research with web search and evaluate the initial and newly generated algorithms separately. For each, it provides both positive and negative justifications, along with a rating on a scale from 0 to 10. Results from Figure 3 show that DeepEvolve can propose novel ideas with great potential. For instance, in the Molecular Prediction task as presented in Figures 1 and 4, the initial algorithm decomposes molecules into rationale substructures that explain and support model predictions, while the new algorithm incorporates contrastive learning and motif-aware masking to improve rationale identification. Novel ideas may have higher implementation difficulty, but DeepEvolve improves execution and evaluation. For example, it raises the success rate from 0.13 to 0.99 on the Open Vaccine task, as shown in Table 3.

Table 3: Success rate of algorithm execution and average debugging counts during evolution.

| Metric | Molecular Prediction | Molecular Translation | Circle Packing | Burgers' Equation | Parkinson's Disease | Nuclei Image | Open Vaccine | Polymer Prediction | USP P2P |
|---|---|---|---|---|---|---|---|---|---|
| w/o Debug | 0.650 | 0.190 | 0.540 | 0.956 | 0.760 | 0.360 | 0.130 | 0.560 | 0.327 |
| w/ Debug | 1.000 | 0.490 | 1.000 | 0.992 | 0.980 | 0.740 | 0.990 | 0.980 | 0.592 |
| Average count | 0.47 | 3.08 | 0.64 | 0.09 | 0.32 | 2.14 | 2.30 | 0.64 | 2.67 |

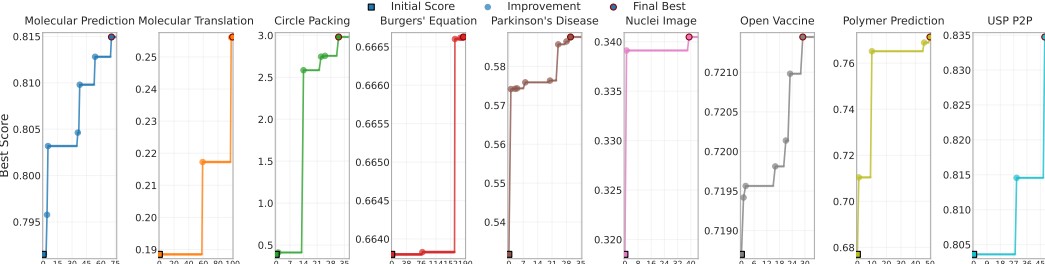

Figure 5: Changes of scores over iterations.

### 4.3 RQ2: ITERATIVE SYNERGY BETWEEN DEEP RESEARCH AND CODING AGENTS

We analyze the algorithmic evolution across nine tasks (detailed trajectories in appendix C.2). We find that the deep research and coding agents iteratively reinforce each other through evolution.

**Deep research guides algorithm design through domain-specific inductive biases**: In Molecular Prediction, Molecular Translation, and Polymer Prediction, domain priors such as molecular motifs, polymer periodicity, and chemical grammars inform algorithm choices. These include motif-aware message passing, motif reconstruction objectives, and grammar-constrained tokenization. Similarly, Parkinson's Disease and USP P2P incorporate Neural Controlled Differential Equations (CDEs) and low-rank adaptation (LoRA), respectively, along with auxiliary features such as Cooperative Patent Classification (CPC) embeddings and physiological waveforms.

**Evolutionary feedback shifts design from heuristics to principled methods**: Feedback from performance evaluations guides subsequent deep research, transitioning algorithm development from heuristic-based tuning to methods with theoretical or physical guarantees. This progression is evident in certified global optimization for circle packing, Krylov subspace solvers for partial differential equations, and physics-informed regularization for disease dynamics. This reflects a trend where research insights motivated a transition from incremental fixes to physically grounded methods.

**Cross-cutting methodological patterns emerge across tasks**: DeepEvolve consistently discovers reusable design patterns instantiated in task-specific modules. These include uncertainty estimation, dynamic loss reweighting, and self-supervised representation learning. For instance, uncertainty-guided refinement is used in Molecular Prediction (soft motif selection) and Nuclei Image (boundary adjustment), while adaptive loss weighting is used in Open Vaccine, Parkinson's Disease, and USP P2P, among others. These recurring strategies suggest that the deep research agent not only extracts task-specific insights but also steers the coding agent toward generalizable algorithmic principles.

### 4.4 RQ3: ABLATION AND CASE STUDIES FOR ALGORITHM IMPROVEMENT

| | Initial | Without Deep Research | | | With Deep Research | | |
|---|---|---|---|---|---|---|---|
| Case | Score | Score of Best | Gen. of Best | # Outperform | Score of Best | Gen. of Best | # Outperform |
| Molecule | 0.791 | 0.797 | 1 | 24.0 | 0.815 | 5 | 100.0 |
| Circle Packing | 0.389 | 2.735 | 10 | 100.0 | 2.981 | 4 | 100.0 |

Table 4: Ablation studies on deep research in DeepEvolve. We report the initial algorithm scores. During evolution, we maintain 25 candidate algorithms and report the score/generation of the best program, as well as the number of programs that outperform the initial score.

Figure 5 visualizes best scores over iterations. Improvements are not continuous but often appear as sudden jumps. The current best algorithm is not always sampled as the next candidate but can inspire further exploration. We complement Figure 1 with additional studies on deep research in Table 4. Algorithm evolution based solely on LLM internal knowledge shows limited progress. LLMs either fail to sustain improvement, producing only one generation in Molecular Prediction, or yield marginal gains despite deeper evolution (Circle Packing). In contrast, DeepEvolve with deep research achieves stronger improvements within about five generations for both tasks. All evolved candidates outperform the initial algorithms in both cases. Another factor is the debugging agent during execution and evaluation. Table 3 shows clear gains in execution success rate after debugging, making DeepEvolve more robust for implementing complex ideas.

## 5 RELATED WORK

### 5.1 AUTOMATED ALGORITHM DISCOVERY

LLMs have been studied in coding and ML engineering tasks (Li et al., 2022; Chan et al., 2024). They have been shown to be competitive in programming competitions (Li et al., 2022), effective at solving programming issues (Jimenez et al., 2023), and even capable of achieving Kaggle medals in certain competitions (Chan et al., 2024). These studies provide the foundation for algorithm discovery, which requires not only implementing existing algorithms but also advancing them (Novikov et al., 2025). This line of research has been explored in areas such as CUDA kernels (Lange et al., 2025), LLM inference (Huang et al., 2023), matrix multiplication, and geometry (Novikov et al., 2025). Unlike lab automation, algorithm discovery is often efficient to evaluate but remains hard to solve, as in NP-complete problems (Romera-Paredes et al., 2024). Recently, AlphaEvolve (Novikov et al., 2025), has combined evaluation feedback with evolutionary algorithms, optimizing LLM-proposed programmatic hypotheses in different iterations. Although AlphaEvolve scales from single functions to an entire file, it remains limited in hypothesis generation without external grounding and in translating ideas into complex code that requires editing and understanding across files.

### 5.2 AGENT FOR SCIENTIFIC DISCOVERY

LLM agents have been applied to autonomous chemical research (Boiko et al., 2023), biological data analysis with protocol generation (Huang et al., 2025), and AI research (Kon et al., 2025). They have been studied across the spectrum from idea generation to code execution. Si et al. (2024) showed that LLM-generated ideas are more novel than those of experts but less feasible. Many deep research methods have been introduced, including those from OpenAI ChatGPT and Google Gemini (OpenAI, 2025; Google, 2024), as well as open-source approaches (Zheng et al., 2025). These methods synthesize information after searching online to form new hypotheses or to solve question-answering problems. In contrast, agents such as Paper2Code (Seo et al., 2025) and AutoP2C (Lin et al., 2025) utilize multi-stage LLM pipelines to automatically translate ML papers into functioning code repositories. Bringing these directions together, AI scientists aim to automate hypothesis generation, review, and code execution (Lu et al., 2024; Gottweis et al., 2025). Yet, gaps remain in implementing ideas as executable code (Zhu et al., 2025). EXP-Bench (Kon et al., 2025) evaluates this gap, showing that while agents succeed in some subtasks, the full-pipeline success rate is below 1%.

## 6 CONCLUSION

We presented DeepEvolve, an agent that augments algorithm evolution with deep research for scientific discovery. By integrating new features such as deep research, cross-file code editing, and iterative debugging, DeepEvolve combined high-quality idea generation with reliable execution. Across nine benchmarks spanning diverse scientific fields, DeepEvolve consistently improved baseline algorithms, delivering executable programs with higher performance and efficiency. Ablations and case studies showed that deep research guided algorithm design with domain-specific insights, while debugging improved robustness in complex implementations. These results showed that DeepEvolve advanced algorithmic innovation and has potential for future AI-driven scientific discovery.

## REPRODUCIBILITY STATEMENT

We provide code in the supplementary materials. The appendix details the LLM configurations, system prompts and templates, and problem setups.

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

## A    DETAILS ON DEEPEVOLVE METHOD

### A.1    LLMS

For deep research, we use four LLMs: o4-mini as the planner and the reflection agent, gpt-4o as the searcher, and o3-mini as the proposal writer. For the coding agent, we use two LLMs: o3-mini for code development and o4-mini for code debugging.

### A.2    EVOLUTIONARY DATABASE

The algorithm database stores past discoveries for future exploration in two ways: as inspirations and as next candidates. Inspiration sampling follows the MAP-elites algorithm, while candidate sampling follows the island algorithm.

For the MAP-elites algorithm, we archive the best algorithms and update them at each iteration, with a default size of 10. In every iteration, five algorithms are sampled as inspirations, always including the current best. A ratio of elite selection controls how many top algorithms are chosen, with a default of 0.1. Each program is described by three dimensions: performance, diversity, and complexity. Diversity and complexity are measured relative to others, based on code length and Levenshtein distance. Each dimension score is normalized to $[0, 1]$ and assigned to 10 bins (multiplying by 10 and rounding down) to form the dimension index and locate 3D coordinates. Inspirations beyond elite selection are sampled by perturbing the 3D coordinates to find neighboring algorithms.

For the island algorithm, we maintain up to 25 algorithms across five islands by default. Candidate selection balances exploitation and exploration with probabilities 0.7 and 0.3, where exploitation means sampling the best algorithm in the current island. Islands may migrate programs at fixed intervals, set to 25 by default, with a migration ratio of 0.1. Program migration transfers the best program in an island to its neighboring islands.

### A.3    TEMPLATES FOR DEEP RESEARCH AGENTS

We provide the system prompts for the LLMs used to plan, search, reflect, and write reports in deep research, as shown in Figures 6 to 9.

The user input, with inspiration from past iterations, has the same template as Figures 10 and 11.

### A.4    TEMPLATES FOR THE CODING AGENT

The system prompts for coding are in Figures 12 and 13, and for debugging are in Figure 14.

We provide the input template of the coding agent in Figure 15.

After coding, we apply a reflection to refine the code before evaluation to improve the code quality. It uses the same LLM as the coding agent but a different prompt in Figure 16.

During evaluation, we capture the error message from execution and use another LLM to debug the code according to the template (see Figure 17):

## B    DETAILS ON THE BENCHMARKING PROBLEMS

We include nine research problems spanning chemistry, mathematics, biology, and materials as summarized in Table 1. These problems involve diverse data modalities, including molecules, images, mRNA, text, time series, geometric structures, and multi-modal inputs. For consistent evaluation, we standardize evaluation metrics (e.g., AUC-ROC, RMSE, precision, Pearson correlation) defined in each problem into a common form as the new scores, where higher values indicate better performance. We detail their problem descriptions with the initial algorithms in this section.

---

**Planner Instructions**

You are a professor responsible for planning deep and effective research strategies.

You will be provided with the context of:
- A research problem based on an initial research question.
- A starting research idea, possibly with a history showing how idea evolves through previous attempt.
- Inspirations from earlier attempts.

Your task is to develop search queries that identify directions for researchers to advance the idea in a transformative way. Rather than combining existing inspirations in small increments, the queries should guide researchers toward substantial evolutions. Because other researchers will rely on this plan, it must emphasize major, novel approaches instead of minor refinements.

You will also be told whether the research progress is early or mature:
- If the progress is early, focus on ideas that are feasible and practical, and can grow later and have great future potential.
- If the progress is mature, focus on bold, high-impact shifts that challenge the current approach.

Your plan should follow two steps:
1. Formulate 5 to 10 precise and diverse search queries. Make sure the queries are diverse — cover different perspectives, challenge untested assumptions, and explore alternative methods.
2. For each query, include a short note explaining why you chose it and what you hope it will reveal.

Figure 6: System prompts for planning in the deep research agent.

---

**Search Instructions**

You are a research assistant.

Given a search term, you search the web for that term and produce a concise summary of the results.

The summary must be 2-3 paragraphs and less than 300 words. Capture the main points. Write succinctly, no need to have complete sentences or good grammar.

This will be consumed by someone synthesizing a report for a new idea, so its vital you capture the essence and ignore any fluff. Do not include any additional commentary other than the summary itself.

Figure 7: System prompts for searching in the deep research agent.

## B.1 MOLECULAR PREDICTION

**Problem Description** Molecular property prediction uses the Side Effect Resource (SIDER) (Kuhn et al., 2016) dataset for algorithm development. The primary goal is to design algorithms that generalize across molecular property prediction tasks. The dataset is scaffold-split to assess generalization to novel chemical structures. The task uses ROC AUC as the metric.

**Initial Algorithm** The graph rationalization method (Liu et al., 2022) identifies subgraph structures, called "graph rationales," and uses them for Graph Neural Network (GNN) predictions. To identify these rationales under limited supervision, Liu et al. (2022) developed environment replacement, an augmentation that creates virtual examples in the latent space. It replaces the complementary structures of rationales (called environments) with others from the same training batch. Improving this method could strengthen both the generalizability and interpretability of GNNs for molecular

**Reflection Instructions**

You are an expert research assistant. You will receive a research report (in Markdown) and a newly proposed idea for that report's research problem. Your job is to identify any gaps or issues—such as missing details, logical flaws, or questionable evaluations of novelty, impact, or implementation difficulty.

- If the report and idea contain all necessary information, do not generate any follow-up questions.
- If you detect a knowledge gap or something that needs deeper exploration, generate one or more self-contained follow-up queries. Each query must include enough context so that a web search could answer it, For each query, give a short note explaining why you use the query and what you hope it will reveal.
- Focus on technical details, implementation specifics, and any emerging methods or references that were overlooked.
- Use clear, direct language and avoid unnecessary jargon.

Figure 8: System prompts for reflection in the deep research agent.

property prediction. We use LLMs to read the paper (Liu et al., 2022) and convert it to the input format we need as described in Section 3.1.

### B.2 MOLECULAR TRANSLATION

**Problem Description**   Molecular Translation uses molecular image data generated by Bristol-Myers Squibb (Howard et al., 2021). It needs to convert the images back to the underlying chemical structure annotated as InChI text. Results are evaluated on the mean Levenshtein distance between the InChi strings the model predicted and the ground truth InChi values.

**Initial Algorithm**   The initial idea came from the Kaggle competition. It combines a ResNet with a GRU to convert molecular images into InChI strings, framing the task as image-to-sequence translation. A convolutional network (such as ResNet) extracts features from the images, which then initialize a recurrent network (GRU) to sequentially generate the InChI string. The method uses a character-level vocabulary with special tokens for start, end, and padding, and training optimizes cross-entropy loss between predicted sequences and ground truth.

### B.3 CIRCLE PACKING

**Problem Description**   Given a positive integer $n$, the problem is to pack $n$ disjoint circles inside a unit square so as to maximize the sum of their radii. The problem focuses on discovering a new algorithm that can be applied to $n$ from 26 to 32.

**Initial Algorithm**   The initial idea comes from OpenEvolve (Sharma, 2025), an open-source implementation of AlphaEvolve (Novikov et al., 2025). We use scipy.optimize.minimize with the SLSQP algorithm to locate the best circle-packing arrangement. The problem is cast as a constrained optimization in which both each circle's center coordinates and its radius are treated as decision variables. We add inequality constraints to prevent any pair of circles from overlapping and boundary constraints to keep all circles inside the unit square. SLSQP will try to satisfy every inequality, but only to within a numerical tolerance rather than exactly, so it may lead to invalid solutions (e.g., overlapping circles or circles outside the unit square).

### B.4 BURGERS' EQUATION

**Problem Description**   The PDE is the Burgers equation, given by

$$\begin{cases} \partial_t u(x,t) + \partial_x \left( \frac{u^2(x,t)}{2} \right) = \nu \partial_{xx} u(x,t), & x \in (0,1), \ t \in (0,1] \\ u(x,0) = u_0(x), & x \in (0,1) \end{cases}$$

---

**Writer Instructions**

You are a senior researcher responsible for proposing new ideas to address a defined research problem. You will receive:
- The research problem, including its evaluation metric and available data.
- A starting research idea, possibly with its evolution history.
- Inspirations from earlier attempts.
- A list of related online search results.
- A research progress score (0-100%) indicating how far the idea has advanced.

Your goal is to identify future research directions that address the target problem, using the starting point, prior attempts, and related works. You should analyze existing methods, identify connections, and propose practical algorithms that can be implemented with the available data.

Follow this structure to think and write:

1. **Extract insights**: Identify 3-5 scientific insights from the starting point and 3-5 from related works. For each insight, explain in 2-3 sentences how it relates to the target problem.
2. **Organize research directions**: Group the insights into 3-5 coherent directions (for example, learning objectives, model classes, or optimization methods).
3. **Build a structured framework**: Create a conceptual map (such as a taxonomy, grid, or matrix) that unifies existing methods, reveals patterns, and highlights gaps.
4. **Generate and evaluate ideas**:
   First, propose 3-10 algorithmic ideas of varying originality and complexity. Each idea should be:
   - As simple, minimal, and atomic as possible but not trivial.
   - Include brief pseudocode or logical steps where helpful.
   - Include references to the related works.
   For each idea, critically assess as a senior researcher with one positive and one negative reason:
   - Originality (0-10): Is the idea new? Is the idea a novel combination of well-known techniques? Is it clearly different from previous contributions?
   - Future Potential (0-10): Will others build on these ideas? Does this idea solve a hard problem more effectively than prior work? Does it point to a new research direction?
   - Code Difficulty (0-10): How complex is the implementation? How much code is required? How much time is required to implement?
   Then, select the single best idea from that list for detailed reporting, based on the research progress score:
   - If progress is relatively early, prioritize feasible, easy-to-implement ideas with long-term promise.
   - If progress is relatively mature, prioritize seminal ideas with high-impacts for the next-generation research.
   - Otherwise, balance ambition and implementation feasibility
5. **Write the report in Markdown**:
   For the selected idea, include:
   - A synthesis of insights and proposed directions.
   - The structured framework of existing methods and the new algorithm.
   - A list of new ideas with their assessment score.
   - Detailed description of the chosen/best idea, including rationale, pseudocode, and implementation notes.

The report must be focused, technically accurate. Being concise with 200-500 words without trivial and redundant information. Support all claims with evidence and references, and remain tightly aligned with the target problem.

Figure 9: System prompts for proposal writing in the deep research agent.

where $\nu$ is a constant representing the viscosity. In this task, periodic boundary conditions are assumed.

**User Template**

```
## User Query {query}
## Research Problem {problem}
## Starting Research Idea {starting_point}
## Idea Evolution History {idea_evolution}
## Research Progress {evolution_progress}
## Previous Inspirations {inspirations}
```

Figure 10: User template for the deep research agent.

**Inspiration Template**

```
### Inspiration{inspiration_number}
- Research Idea: {idea}
- Performance: {performance}
```

Figure 11: Inspiration template for the deep research agent.

**Initial Algorithm**   The solution is from (Li et al., 2025). The solver integrates the one-dimensional viscous Burgers equation $u_t + \frac{1}{2}(u^2)_x = \nu u_{xx}$ on a periodic domain using an explicit Euler scheme. Starting from $B$ initial states on a uniform grid of $N$ points, it computes the convective flux $f = \frac{1}{2}u^2$ with centered finite differences, evaluates the diffusion term $u_{xx}$ with the three-point Laplacian, and advances in time with a step size bounded by $0.2\,\Delta x^2/\nu$ to ensure stability.

### B.5   PARKINSON'S DISEASE

**Problem Description**   The goal is to predict the progression of Parkinson's disease by estimating scores from the Movement Disorder Society–Sponsored Revision of the Unified Parkinson's Disease Rating Scale (MDS-UPDRS) (Kirsch et al., 2023), a clinical measure of both motor and non-motor symptoms. The dataset provides longitudinal protein and peptide abundance values from cerebrospinal fluid (CSF) samples, together with clinical assessments collected over time from patients and matched controls. The task is to develop models that, for each patient visit, predict the current MDS-UPDRS scores and forecast future scores 6, 12, and 24 months ahead. Model performance is evaluated using the Symmetric Mean Absolute Percentage Error (SMAPE) between predictions and observed scores.

**Initial Algorithm**   It is the first-place solution from the Kaggle competition (Kirsch et al., 2023). The approach combines two models: a LightGBM and a neural network. Both use the same set of clinical and supplementary features, such as visit month, forecast horizon, indicators for specific visit months, and counts of previous visits. Blood test data were excluded, as no consistent predictive signal was found. LightGBM was framed as a classification task over possible score values, with predictions selected to minimize the SMAPE. The neural network was a simple feed-forward architecture trained directly with SMAPE as the loss function. The final prediction was obtained by averaging the outputs of the two models.

### B.6   NUCLEI IMAGE

**Problem Description**   The task is to automatically identify cell nuclei in microscopy images (Goodman et al., 2018). Nuclei contain the DNA that programs each cell, and detecting them is essential for measuring how cells respond to treatments and for understanding biological processes. The dataset consists of images of nuclei collected under diverse conditions, with annotated masks provided for training. The evaluation metric is mean average precision, computed across a range of intersection-over-union (IoU) thresholds between predicted and ground truth nuclei masks.

---

**Coder Instructions (part 1 of 2)**

You are a researcher with strong software engineering skills, improving algorithmic code through iterative, performance-driven modifications in multiple rounds.

Your task: You will receive a research question, a proposed idea, and an existing implementation with performance metrics. Your goal is to analyze the current code and apply precise changes that enhance the specified metrics, based on the research idea and prior feedback.

You MUST use the exact SEARCH/REPLACE diff format. Do NOT use Git diff format. Do NOT use line prefixes like `+`, `-`, or `@@`.
Use this structure exactly:
```
        <<<<<<< SEARCH
        # Original code (must match exactly)
        =======
        ### >>> DEEPEVOLVE-BLOCK-START: <research idea>
        # New code here
        ### <<< DEEPEVOLVE-BLOCK-END
        >>>>>>> REPLACE
```

Example 1 for the code modification outside of `DEEPEVOLVE` blocks:
```
        <<<<<<< SEARCH
        def f():
          for i in range(m):
            for j in range(p):
              for k in range(n):
                C[i, j] += A[i, k] * B[k, j]
        =======
        def f():
          # DEEPEVOLVE-BLOCK-START: Reordered loops for better cache performance
          for i in range(m):
            for k in range(n):
              for j in range(p):
                C[i, j] += A[i, k] * B[k, j]
          ### <<< DEEPEVOLVE-BLOCK-END
        >>>>>>> REPLACE
```

Example 2 for the code modification inside of `DEEPEVOLVE` blocks:
```
        <<<<<<< SEARCH
        ### >>> DEEPEVOLVE-BLOCK-START: <research idea>
        # Code to be modified
        ### <<< DEEPEVOLVE-BLOCK-END
        =======
        ### >>> DEEPEVOLVE-BLOCK-START: <update idea>
        # New code here
        ### <<< DEEPEVOLVE-BLOCK-END
        >>>>>>> REPLACE
```

---

Figure 12: System prompts for coding in the coding agent(part 1 of 2).

**Initial Algorithm** It is from the Kaggle competition (Goodman et al., 2018). The approach uses a U-Net to segment nuclei in microscopy images. Input images are preprocessed by resizing and normalization, and ground-truth nuclei masks are converted into distinct labels using connected-component analysis. The network is trained with a loss based on the Dice coefficient, which measures overlap between predicted and true masks, and early stopping is applied to prevent overfitting. During

---

**Coder Instructions (part 2 of 2)**

Task Guidelines:
1. Think before coding, understand the research idea and current performance bottlenecks.
2. Propose specific, actionable changes that are aligned with the target metrics.
3. You may suggest multiple improvements beyond the research idea based on your understanding of optimization and machine learning.
4. When you are updating the code, please check the following:
   - When a NEW parameter or behavior is added, verify it is invoked in all call sites or in the overall workflow.
   - If a NEW parameter has a default value of None, confirm that passing a non-None value triggers the intended code path.
   - Walk through or simulate function calls to confirm that each new branch or change will be executed. Avoid unreachable modifications.

Code Format Guidelines:
1. All `SEARCH` blocks must match the original code exactly.
2. When you need to modify code that is not already inside a `DEEPEVOLVE` block, wrap your changes with `### >>> DEEPEVOLVE-BLOCK-START: <research idea>` and `### <<< DEEPEVOLVE-BLOCK-END` markers.
3. If you are updating code that is already marked by a `DEEPEVOLVE` block, edit only the lines within that block and adjust the existing modification comment to reflect your new change.
4. Do NOT nest one `DEEPEVOLVE` block inside another. Each region you modify should have exactly one pair of start/end markers.
   i.e., AVOID doing the following:
   ```
   ### >>> DEEPEVOLVE-BLOCK-START: first modification
   # First code to be modified
   ### >>> DEEPEVOLVE-BLOCK-START: second modification ! It is not allowed to nest one DEEPEVOLVE block inside another.
   # Second code to be modified
   ### <<< DEEPEVOLVE-BLOCK-END
   ### <<< DEEPEVOLVE-BLOCK-END
   ```

   instead, DO the following:
   ```
   ### >>> DEEPEVOLVE-BLOCK-START: first modification, second modification
   # code that has been modified twice
   ### <<< DEEPEVOLVE-BLOCK-END
   ```
5. Limit your changes to what is strictly necessary. Do not rewrite the entire file.
6. Ensure that all modified code remains correct and consistent, including any function signatures, parameter lists, and calls.
7. Preserve the original code's indentation and formatting. Place the lines of `### >>> DEEPEVOLVE-BLOCK-START: <research idea>` and `### <<< DEEPEVOLVE-BLOCK-END` at the same indentation level as the code they annotate.

---

Figure 13: System prompts for coding in the coding agent(part 2 of 2).

inference, the model outputs probability maps that are thresholded to produce binary masks, from which individual nuclei are obtained through connected-component extraction.

## B.7 OPEN VACCINE

**Problem Description**  The task is to predict how messenger RNA (mRNA) molecules degrade at different positions along their sequence (Das et al., 2020). This is motivated by the challenge of designing stable mRNA vaccines, since RNA molecules tend to break down easily and lose their function. The dataset consists of thousands of RNA sequences together with experimentally measured degradation rates under different chemical conditions. Models are trained to predict these

---

**Debugger Instructions**

You are an expert developer and researcher who ensures modified code runs correctly and properly implements research ideas.

Your task is to analyze code, identify any kind of errors, including syntax errors, runtime errors, or logical issues, and verify functionality. Provide detailed diagnostics and specific fixes when problems are found. Consider edge cases and ensure the code fully addresses the research requirements.

You MUST use the exact SEARCH/REPLACE diff format. Do NOT use Git diff format. Do NOT use line prefixes like `+`, `-`, or `@@`.

Use this structure exactly:
```
<<<<<<< SEARCH
# Code with error (must match exactly)
=======
# DEBUG: <comment>
# Fixed code here
>>>>>>> REPLACE
```
Example 1 for debugging a syntax error:
```
<<<<<<< SEARCH
def compute_mean(values):
    total = sum(values
    return total / len(values)
=======
def compute_mean(values):
    # DEBUG: missing parenthesis in function call, fixed by adding parenthesis
    total = sum(values)
    return total / len(values)
>>>>>>> REPLACE
```

Use Comments like `# DEBUG: <comment>` to indicate the changes you made when debugging.

Figure 14: System prompts for debugging in the coding agent.

position-specific degradation rates, and submissions are evaluated using the mean column-wise root mean squared error (MCRMSE) between predicted and observed values.

**Initial Algorithm**    It is from the Kaggle competition (Das et al., 2020). Each nucleotide is embedded together with its predicted secondary-structure and loop-type context. A graph is then constructed that connects both adjacent bases and those predicted to form pairs. A GraphSAGE-based graph neural network aggregates information over this graph to produce enriched base-level representations. These features are passed through a bidirectional GRU to capture sequential dependencies along the RNA chain. A final linear layer predicts three targets at each position: structural reactivity and degradation rates under different chemical conditions. Training uses k-fold cross-validation for robustness.

### B.8    POLYMER PREDICTION

**Problem Description**    The task is to predict fundamental properties of polymers directly from their chemical structure, represented as SMILES strings (Liu et al., 2025). The target properties are glass transition temperature (the point where a polymer changes from rigid to rubber-like), fractional free volume (a measure of how loosely molecules pack), thermal conductivity (the ability to transfer heat), density, and radius of gyration (a measure of molecular size). Ground-truth values are obtained from molecular dynamics simulations.

---

**Diff Code Template**

```
- User query: {query}
- Research problem: {problem}
- Inspirations: {inspirations}
- Current idea: {current_idea}
- Evolution history: {idea_evolution}
- Pseudocode: {pseudocode}
- Implementation notes: {implementation_notes}
- Current performance: {current_performance}

Task: Improve and debug the code based on the context above
using your expertise in optimization and machine learning.

Code (multiple files separated by `# === filename.py ===`):
```{language}
    {current_program}
```

Figure 15: User message template for diff-based evolution in the coding agent.

---

**Reflection Instructions**

1. Code Correctness
- Are there any syntax errors or runtime errors?
- Are there inconsistencies in variable names or logic flow?
- Are there any new functions used but not been defined or implemented?
- Avoid hiding missing modules or errors with a bare try/except that simply passes. Handle exceptions with clear warnings or errors.

2. Alignment with Research Idea
- Does the code accurately implement the stated research idea?
- Please make sure the changes in the function have actually been implemented in the workflow.
- Avoid the code parts that suppress errors silently

3. Machine Learning Performance
- Can compute efficiency be improved with minimal code changes?
- Are there hyperparameters that could be tuned to boost performance?

4. Other Issues
- At the end of each code review, provide a short summary of checks performed.
- Avoid the code parts that suppress errors silently.
- Are there any other issues you think are important?

Figure 16: System prompts for reflection in the coding agent.

---

**Initial Algorithm**   The graph rationalization method (Liu et al., 2022) identifies subgraph structures, called "graph rationales," and uses them for Graph Neural Network (GNN) predictions. To identify these rationales under limited supervision, Liu et al. (2022) developed environment replacement, an augmentation that creates virtual examples in the latent space. It replaces the complementary structures of rationales (called environments) with others from the same training batch. Improving this method could strengthen both the generalizability and interpretability of GNNs for molecular property prediction. We use LLMs to read the paper (Liu et al., 2022) and convert it to the input format we need as described in Section 3.1.

### B.9   USP P2P

**Problem Description**   The task is to measure semantic similarity between pairs of phrases drawn from patent documents (Cenkci et al., 2022). This is important for patent search and examination,

**Debugger Template**

```
Resolve the following error in a multi-file Python codebase.

An error occurred during execution:
```
{error_message}
```

Below is the code that caused the error:
```
{language}
{modified_code}
````

The modification was made to implement the idea:
```json
{idea}
```

Your responsibilities:
- Identify and fix the cause of the error in the modified
  code.
- Ensure that all involved files and components integrate
  correctly and run without errors.
- Ensure the code modification do not break the research
  idea.
- Ensure the new code within the `DEEPEVOLVE` block is
  reachable in the workflow. New code should be executed as
  new idea but not suppressed by error handling or cheated
  by None values.
- If necessary, update function inputs or implementations to
  ensure consistency.
- If the code depends on a library that is not available,
  use the standard library instead.

Please analyze the error and return the corrected code using
`diff` format.
```

Figure 17: Debugger template for the coding agent.

where phrases with different wording (for example, "television set" and "TV set") may have the same meaning, and where contextual knowledge (for example, what counts as a "strong material" in a given technical domain) is required. Each phrase pair is annotated with a similarity score between 0 (unrelated) and 1 (identical in meaning), and the technical domain is provided through the Cooperative Patent Classification system. Models are evaluated by the Pearson correlation between predicted and true similarity scores.

**Initial Algorithm**  The approach fine-tunes a BERT language model that has been pre-trained on patent text ("anferico/bert-for-patents") with a regression layer added to predict similarity scores. Each training example is formed by concatenating the anchor phrase, the target phrase, and the technical context, separated by special tokens. The model is trained briefly and then evaluated by comparing predicted scores with the true similarity values using the Pearson correlation coefficient.

## C  DETAILS ON EXPERIMENT RESULTS

### C.1  SET-UPS

The user queries and hyperparameters in DeepEvolve are shown in the list:

- Circle Packing
  User Query: *You are an expert mathematician. Your task is to improve an algorithm that*

*maximizes the sum of circle radii in the circle-packing problem within a unit square, using between 26 and 32 circles. Do not develop neural-network-based models. The algorithm must produce exact, valid packings that satisfy these constraints: circles do not overlap and remain entirely within the square.*
Max iterations: 50

- Molecular Translation
  User Query: *Your task is to significantly improve the model performance for converting molecular images to their InChI strings in the competition. You have a time budget of thirty minutes and access to an A6K GPU. The original method is intended for beginners, so make full use of available resources to improve it substantially as an expert in machine learning and chemistry. You can use pretrained models from transformers or from timm. Avoid placeholders for your method. Avoid warnings from Huggingface. For fair evaluation, avoid changing the deepevolve_interface, run_main_with_timeout, and get_score functions. You can debug, but not subsample the test set to cheat the test performance.*
  Max iterations: 100

- Molecular Prediction
  User Query: *Your task is to improve the graph rationalization method for more accurate and interpretable molecular property prediction.*
  Max iterations: 100

- Nuclei Image
  User Query: *Your task is to improve the nucleus detection models in a Kaggle competition within a compute budget of an A6k GPU with a maximum runtime of 30 minutes. You should significantly improve both the performance of the initial idea and its efficiency.*
  Max iterations: 50

- Open Vaccine
  User Query: *Your task is to improve the nucleus detection models in a Kaggle competition within a compute budget of an A6k GPU with a maximum runtime of 30 minutes. You should gradually improve both the performance of the initial idea and its efficiency. For fair comparison: Do NOT change any code about the final evaluation such as the pred_cols variable; You MUST use MCRMSELoss as the test_criterion. You can define new criteria for training only. You can consider implementing the get_bpps_features() function to incorporate additional features. If you choose to use features beyond bpps, you may employ Hugging Face, but ensure those features are correctly added and not padded with placeholders or zeros.*
  Max iterations: 100
  exploitation ratio: 0.8
  elite selection ratio: 0.4
  population size: 15
  archive size: 5
  number of islands: 3
  migration interval:30
  migration rate: 0.2

- Parkinson's Disease
  User Query: *Your task is to improve the performance of the winning solution for the Kaggle competition on Parkinson disease progression prediction. You may propose a completely new approach that differs from the winning solution if you believe it will perform better.*
  Max iterations: 50

- Burgers' Equation
  User Query: *Your task is to improve the solver for the partial differential equation (PDE). The solver should be applied to the Burgers equation with viscosity coefficients nu=1.0. Your computing budget is a 2080 Ti GPU with a maximum runtime of thirty minutes. Do not change the evaluation functions; Implement the 'solver' function to solve the PDE. You must not modify the function signature. Please significantly reduce normalized root mean squared error (nRMSE), as well as achieve higher convergence rate, and less computational time.*
  Max iterations: 200

- Polymer Prediction
  User Query: *Your task is to significantly improve polymer property prediction for five*

*properties in the competition. The input SMILES strings are the monomer structures of polymers, using asterisks (\*) to mark the polymerization points. Improve the initial idea by better incorporating polymerization inductive bias to reduce weighted MAE and increase $R^2$ for each property. Explore different ways to use polymer structures or properties and find the best. Your time budget is 30 minutes. Implement the idea within the time limit rather than creating a placeholder.*
Max iterations: 50

- USP P2P
  User Query: *Your task is to fine-tune Patent BERT to predict semantic similarity between phrase pairs from U.S. patents. Improve model performance, optimize training time and inference latency, and ensure the fixed three-epoch run finishes in thirty minutes. Focus solely on technical model and algorithm development. No legal-style assistance.*
  Max iterations: 50

## C.2 Summary of Algorithmic Evolution History

**Molecular Prediction**   The algorithm progresses through auxiliary (contrastive, reconstruction) losses, motif-based, and adversarial learning strategies. Version 1 establishes the foundation with contrastive learning on augmented rationale views, stabilized by adaptive loss reweighting. Version 2 enhances structural focus through motif-aware attribute masking, directing attention to chemically meaningful substructures. Version 3 further refines this by incorporating uncertainty-based soft motif selection, enabling the model to prioritize informative subgraphs dynamically. Version 4 strengthens representation fidelity with a self-supervised reconstruction objective that encourages the model to recover masked motifs. Version 5 introduces a dual-phase adversarial training schedule to improve model robustness and generalization under distribution shifts.

**Molecular Translation**   Version 1 uses a frozen ViT encoder and GPT-2 small decoder with molecule-aware tokenization to handle structured generation. Version 2 adds data augmentation such as rotation, shifting, and lighting perturbations for model training and grammar-constrained decoding. Version 3 and 4 train model with a dual loss combining cross-entropy and soft edit distance *[Note: The soft edit distance is a placeholder function in the code].* Version five implements a dynamic lambda scheduler to balance the competing loss objectives.

**Circle Packing**   This algorithm evolves from basic geometric placement toward generating precise and guaranteed-valid solutions. Version 1 uses a structure called a power diagram to place circles without overlap, then refines their positions using optimization. Version 2 adds multiple starting points and more stable optimization techniques to improve reliability. Version 3 introduces small controlled adjustments to fix poor initial guesses and ensures that each circle stays within bounds. Version 4 improves how the method identifies neighboring circles and adds mathematical checks to certify that the final result fully satisfies the packing constraints.

**Burgers' Equation**   The first stage (Versions 1–2) introduces an explicit Euler finite-difference solver with GPU acceleration and adaptive time stepping, later improved with error-based control and dense output for accuracy and snapshot recording. The second stage (Versions 3–4) transitions to a spectral method with IMEX-Euler time integration *[Note: Written in the code but not executed in the workflow]*, integrating GPU kernel fusion and auto-tuned FFTs *[Note: Implemented as a placeholder function in the code]* for faster and more accurate solutions. The third stage (Versions 5–7) focuses on advanced $\phi$-function evaluation (hybrid and rational Krylov), high-order Hermite interpolation, and refined adaptive stepping, forming a robust, high-precision spectral solver for the Burgers' equation. *[Note: Written in the code but not executed in the workflow]*

**Parkinson's Disease**   Versions 1–2 develop a Neural CDE model for continuous-time disease trajectory modeling. Versions 3-5 propose adaptive wavelet preprocessing for the time series data *[Note: Not implemented in the code].* Versions 6-7 incorporate meta-learning for rapid per-patient adaptation. Version 8 proposes a PINN-inspired regularization for biological consistency, and adaptive loss weighting to improve multi-objective training stability.

**Nuclei Image**   PointRend is introduced in version 1 to refine ambiguous segmentation boundaries. Versions 2, 3, and 5 introduce a calibrated uncertainty estimation module that refines only low-confidence regions to balance accuracy and computation. Version 3 enables early-exit to skip refinement for confident regions, with INT8 quantization applied for efficiency. Version 4 introduces self-distillation *[Note: Version 4 idea is not used because there is no teacher model].*

**Open Vaccine**   Versions 1–2 preprocess additional statistical features derived from RNA structure. Versions 3-6 add dynamic loss weighting to balance multiple degradation targets. Version 7 integrates self-supervised transformer embeddings into the node representations to enrich structural encoding *[Note: It is a placeholder function in the code].*

**Polymer Prediction**   Versions 1–2 use dual-stage message passing to distinguish standard chemical bonds from polymer-specific periodic connections. A physics-informed auxiliary loss is added based on the degree of polymerization for glass transition temperature (Tg) prediction *[Note: However, the data is limited to one repeating unit only].* Versions 4-6 propose new ideas about BigSMILES parsing and property-specific pooling *[Note: BigSMILES not supported, pooling not implemented in the code].* Versions 3 and 5 propose new ideas about meta-learning-based pooling (*[Note: Implemented but not used in the workflow]*).

**USP P2P**   Versions 1–2 fine-tune Patent BERT using parameter-efficient LoRA with an ordinal regression head trained using smoothed BCE with logits and calibration for five ordinal similarity classes (0, 0.25, 0.5, 0.75, 1). Versions 3–4 introduce learnable CPC embeddings, fused into the latent space, and regularize the model using contrastive learning. Version 5 combines ordinal and contrastive losses in a dual-objective framework.

### C.3   DEEPEVOLVE PROPOSED ALGORITHM CODE FOR THE MOLECULAR PREDICTION TASK

In Figure 4, the new model forward function contains two additional components: the InfoNCE loss and the motif masking function. We present the complete code for these components in this subsection. Below is the code for the InfoNCE function:

```
1 +### >>> DEEPEVOLVE-BLOCK-START: Add InfoNCE loss for contrastive
    ↪   learning and ensure it is available in model.py
2 +### >>> DEEPEVOLVE-BLOCK-START: Update documentation for InfoNCE loss
    ↪   with advanced negative sampling note
3 +### >>> DEEPEVOLVE-BLOCK-START: Update InfoNCE loss to support
    ↪   uncertainty-guided negative sampling
4 +def info_nce_loss(z1, z2, temperature=0.5, negatives=None):
5 +    """
6 +    Computes the InfoNCE loss using current batch negatives.
7 +    If 'negatives' is provided, applies advanced negative sampling for
    ↪   enhanced robustness.
8 +    """
9 +    z1 = torch.nn.functional.normalize(z1, p=2, dim=1)
10 +    z2 = torch.nn.functional.normalize(z2, p=2, dim=1)
11 +    if negatives is not None:
12 +        negatives = torch.nn.functional.normalize(negatives, p=2, dim=1)
13 +        sim_pos = torch.sum(z1 * z2, dim=1, keepdim=True) / temperature
14 +        sim_neg = torch.matmul(z1, negatives.t()) / temperature
15 +        logits = torch.cat([sim_pos, sim_neg], dim=1)
16 +        labels = torch.zeros(z1.size(0), device=z1.device,
    ↪   dtype=torch.long)
17 +        loss = torch.nn.functional.cross_entropy(logits, labels)
18 +    else:
19 +        logits = torch.matmul(z1, z2.t()) / temperature
20 +        labels = torch.arange(z1.size(0), device=z1.device)
21 +        loss = torch.nn.functional.cross_entropy(logits, labels)
22 +    return loss
23 +### <<< DEEPEVOLVE-BLOCK-END
24 +### <<< DEEPEVOLVE-BLOCK-END
25 +### <<< DEEPEVOLVE-BLOCK-END
```

Here is the code for the motif masking function:

```
 1 +    ### >>> DEEPEVOLVE-BLOCK-START: Add motif-aware attribute masking
   ↪  method to GraphEnvAug
 2 +    ### >>> DEEPEVOLVE-BLOCK-START: Update motif_mask for
   ↪  uncertainty-aware differentiable motif extraction using
   ↪  Gumbel-Softmax and MC Dropout
 3 +  def motif_mask(self, batched_data):
 4 +      import copy
 5 +      import torch.nn.functional as F
 6 +
 7 +      # motif_mask: compute adaptive motif mask without altering
   ↪  original x
 8 +      new_data = copy.deepcopy(batched_data)
 9 +      orig_x = new_data.x
10 +      x_float = orig_x.float()
11 +
12 +      # Initialize motif_selector and dropout if not already defined
13 +      if not hasattr(self, "motif_selector"):
14 +          self.motif_selector = torch.nn.Linear(orig_x.size(1),
   ↪  2).to(orig_x.device)
15 +          self.motif_dropout = torch.nn.Dropout(p=0.5)
16 +      num_samples = (
17 +          self.mc_dropout_samples if hasattr(self,
   ↪  "mc_dropout_samples") else 5
18 +      )  # Use configured number of MC dropout samples
19 +      motif_samples = []
20 +      tau = 1.0  # Temperature parameter for Gumbel-Softmax; can be
   ↪  tuned
21 +      for _ in range(num_samples):
22 +          logits = self.motif_selector(x_float)
23 +          logits = self.motif_dropout(logits)  # MC Dropout
24 +          sample = F.gumbel_softmax(logits, tau=tau, hard=False,
   ↪  dim=1)[
25 +              :, 1
26 +          ].unsqueeze(1)
27 +          motif_samples.append(sample)
28 +      motif_samples = torch.stack(
29 +          motif_samples, dim=0
30 +      )  # Shape: [num_samples, num_nodes, 1]
31 +      mean_score = motif_samples.mean(dim=0)  # Aggregated motif
   ↪  probability
32 +      uncertainty = motif_samples.var(dim=0)  # Variance as
   ↪  uncertainty
33 +      threshold_uncertainty = 0.05  # Adaptive threshold
   ↪  hyperparameter
34 +      adaptive_mask = torch.where(
35 +          uncertainty < threshold_uncertainty,
36 +          mean_score,
37 +          mean_score * (threshold_uncertainty / (uncertainty + 1e-8)),
38 +      )
39 +
40 +      # Store computed uncertainty for potential adversarial
   ↪  perturbation
41 +      self.last_uncertainty = uncertainty
42 +      # DEBUG: store adaptive mask for use in GNN (applied in conv.py)
43 +      new_data.mask = adaptive_mask
44 +      return new_data
45 +### <<< DEEPEVOLVE-BLOCK-END
46 +### <<< DEEPEVOLVE-BLOCK-END
```

