# OpenReview forum: "Scientific Algorithm Discovery by Augmenting AlphaEvolve with Deep Research"
_ICLR.cc/2026/Conference — Submitted to ICLR 2026_

### Official Review · Reviewer_w3Ad · 2025-10-26

**Soundness:** 3
**Presentation:** 3
**Contribution:** 2
**Rating:** 4
**Confidence:** 3

**Summary:**

The paper proposes DeepEvolve, an agent that couples deep research (planning, web retrieval, synthesis) with evolutionary code search (cross-file edits, automated debugging, and selection) to discover improved algorithms. The system instantiates six modules—plan, search, write, code, evaluation, and evolutionary selection—and is applied to nine problems spanning chemistry, mathematics, biology, materials, and patents. Reported results indicate improvements over each task’s initial algorithm and suggest that adding deep research accelerates progress relative to pure evolution.

**Strengths:**

- Clear problem statement, motivation, and pipeline design. The paper formalizes algorithm discovery with a textualization function and an update operator applied over K rounds, and it specifies the six-module workflow with concrete roles for each agent and a coupled evaluation loop.
- Experiments show the effectiveness of the method to some extent. The study covers nine tasks with different modalities and domains, along with metric standardization to the “new score”.
- The paper also emphasizes a budget of compute/iteration, which is a valuable consideration for realistic, time-boxed algorithm search.
- The related works are sufficient there.
- The code is provided in the supplementary materials.

**Weaknesses:**

- I did not fully get the novelty/technical novelty of the pipeline, as the high-level idea ("a pipeline that couples "deep research" with "code evolution" in a closed loop") has been shown in previous works in different domains (chemistry, physics). The difference can be more like adding a constraint for allowing agents to continuously produce "runnable" new algorithms under tight budgets?
- Baselines not sufficiently strong or comprehensive... Main comparisons largely pit DeepEvolve against each task’s initial algorithm. There is no systematic, head-to-head evaluation against strong/SOTA task solvers or a robust AlphaEvolve-only / pure-evolution baseline across all tasks. As a result, it is difficult to attribute gains to deep research versus implementation specifics or easier starting points.

- Also, the ablation evidence focuses on a subset of tasks; cross-domain claims (“works across nine areas”) are under-supported without broader, per-domain ablations (e.g., research-only vs. code-only, search breadth/depth sweeps, no-debugger controls).

- Subjective evaluation via LLM-as-a-judge. Originality, future potential, and difficulty are assessed by an LLM without human calibration or agreement checks - these scores should be treated as qualitative at best.

**Questions:**

Besides the questions mentioned above:

- For each task with a transformed “new score,” can the authors report the native metric alongside statistical uncertainty (e.g., mean±std over seeds) and justify the specific weighting schemes in Table 1 to clarify sensitivity to the aggregation?
- How many independent runs were performed per candidate per iteration, and what is the variance of the final selected algorithms under re-runs with different random seeds and retrieval states?
- Can the authors add intermediate ablations that vary the number of planning/reflection cycles or the breadth of retrieved sources to quantify how research depth influences the performance?

**Details Of Ethics Concerns:**

/

---

### Official Review · Reviewer_oCHe · 2025-10-31

**Soundness:** 2
**Presentation:** 3
**Contribution:** 3
**Rating:** 4
**Confidence:** 4

**Summary:**

This paper presents DeepEvolve, an AI-driven framework for scientific algorithm discovery that integrates deep research (knowledge retrieval and synthesis) with algorithm evolution (iterative code generation and optimization).The core idea is to bridge the gap between unguided algorithm evolution (e.g., AlphaEvolve) and ungrounded deep research by forming a closed-loop system consisting of six interacting modules: planning, searching, writing, coding, evaluation, and evolutionary database management. DeepEvolve employs external information retrieval (from sources like arXiv or PubMed), cross-file code editing, automated debugging, and an island-based evolutionary database with MAP-Elites sampling. The framework is tested on nine benchmark problems across chemistry, mathematics, biology, materials, and patent analysis, showing consistent improvements over baselines and AlphaEvolve-style systems.
In essence, DeepEvolve proposes a meta-level black-box optimization system that learns to generate, refine, and validate algorithms across diverse scientific domains.

**Strengths:**

1. DeepEvolve unifies deep research, implementation, and evolutionary selection, achieving a closed-loop discovery process.
2. Multi-agent orchestration, debugging, and cross-file code reasoning significantly increase executable success rate.
3. Nine tasks across multiple modalities demonstrate domain generality.

**Weaknesses:**

1. Experiments compare only to initial baselines or AlphaEvolve reproductions, omitting recent systems. Without such baselines, the empirical advantage remains unclear.
2. No standard deviations, p-values, or multiple-run averages are reported. Some gains (e.g., +0.39%) may not be significant.
3. MAP-Elites dimensions (Levenshtein distance, length) capture syntactic, not semantic, diversity. Thus “diverse” candidates might differ textually but not algorithmically.
4. Runtime (30 min per GPU) is mentioned but lacks breakdown of LLM calls, token usage, or cost-performance trade-off.
5. While the ablation study in Table 4 provides evidence that the Deep Research component contributes significantly to performance, it remains unclear how other modules within the framework (e.g., Reflection, Debugging, Cross-file Coding, Evolutionary Database) individually affect outcomes.
6. Appendix explicitly states the use of different models (o4-mini, o3-mini, gpt-4o) for distinct agents.However, no experiment examines how this design affects performance or cost.Without such analysis, it’s impossible to know if the framework’s improvements arise from architecture design or simply from stronger LLMs performing key roles.

**Questions:**

1. Could you provide statistical confidence intervals or multiple-seed averages for Table 2?
2. Can you provide a comparison with recent algorithm-discovery agents (e.g., AutoP2C)?
3. Are the newly discovered algorithms generalizable to unseen datasets or only overfit the current benchmarks?
4.Could you provide ablation results isolating the impact of reflection, debugging, and evolutionary database components?
5. How stable is performance if model roles are swapped (e.g., o3-mini as searcher)?

---

### Official Review · Reviewer_hYKd · 2025-11-06

**Soundness:** 1
**Presentation:** 3
**Contribution:** 2
**Rating:** 2
**Confidence:** 3

**Summary:**

The paper extends AlphaEvolve-style algorithm evolution with a deep research agent, that searches the web for relevant information and informs proposals for new algorithms.

The new approach is tested on 9 problems from different scientific areas.

**Strengths:**

The paper presents a novel approach of combing a deep research agent with evolutionary algorithm discovery. This could be significant especially to incorporate very recent knowledge or when the recall abilities of the LM are not sufficient to make full use of the available internet knowledge are.

The paper is clearly written and understandable, though a focus on clearly proving the main point (deep research is helpful for algorithm discovery) would still improve it.

The agent design seems reasonable, however at the current state of ablations and experiments I am unconvinced by the data about how much gain the new design leads to.

**Weaknesses:**

My main issue at this point are the ablations and experiments. The main point of the paper seems to be to demonstrate that adding deep research to a AlphaCode-style evolutionary algorithm discovery leads to better results.

However, the main quantitative results (tab 2) do not compare different algorithm discovery strategies, but only compare improvements over an algorithm baseline.

The LM as a judge analysis of analyzing the solution space in terms of originality, potential, and implementation difficulty is not explained in detail or validated to avoid biases from LM as a judge setups.

The qualitative observations in section 4.3 could be improved by considering alternative explanations for the observed behavior (see questions below).

The ablation in Tab. 4 is great and should be the core piece of evidence for the superiority of the new method: By combining the scores of the agent with and without deep research. However this is only done for a subset of 2 out of the 9 problems and the difference with or without deep research seem to be exaggerated in the overall framing of the paper and in the accompanying text.

I am willing to raise my score if my questions below are answered and/or additional evidence for the gains by adding deep research are added to the paper.

**Questions:**

Could you provide the reasoning for the exact form of the new scores derived from original metrics as shown in Table 1? In the main text you only mention that you want to ensure that "larger means better" (l. 269), but then you could have just taken the negative of all the scores you wanted to reverse. Or the inverse. Don't your improvement values change depending on how you redefine the scores? I was also surprised to see 1-levenshtein distance because levenshtein distance isn't bounded to [0, 1] (unless it's normalized?), so what motivated this choice of transformation?

Table 2: I have trouble contextualizing these score improvements, because I do not know how good the initial implementations are or what the current state of the art of scientific algorithm discovery is. Would it not be a better and fairer comparison, to start with an initial algorithm (doesn't matter how good or bad) and then compare the performance of AlphaEvolve (or an open-source implementation thereof) with the performance of the strategy presented in this paper? For example in line 323 the paper attributes attaining less performance improvement to using a "recent SoTA implementation for the baseline". But if that means that most of the other baselines are not SoTA, how can I understand how good of an improvement e.g., the 666% are (or if it rather tells me something about the initial baseline)?

Fig 3: What's the correlation between the three quantities? Did you perform any validation of the LM as a judge setup? I'm not entirely sure what the input for the LM judge is (is it the code or just the sketch of the code?), but did you look into simple biases like that longer sketches automatically get higher scores? From my own observation, when LMs are tasked to improve something, they almost always make things more complicated, but that doesn't mean better or more performant. So any validations on the statements derived from the LM as a judge outputs would be good. Including the prompts along of an output example in the appendix would also be good. Another potential validation of Fig 3 would possibly be to look at correlations between the "Future potential score" and the actual obtained score (as given in Tab 2).

Also Fig 3: When the baseline algorithm is an algorithm you implemented based on a paper, doesn't that automatically reduce the baseline originality score to 0? So it seems that it's not at all surprising to see big gains in this category for the new algorithm. Similarly (and again that depends on the exact prompts), does implementation difficulty factor in how much code is already available online (I'm asking because the paper said it's a web-search enabled deep research agent). Because if does, then again there might be a big bias here.


line 402-408: How do we distinguish between that being from deep research vs innate knowledge of the model?

Section 4.3 overall seems very qualitative, it is not clear to me how the conclusions were reached (was this based on a subset of trajectories that was read? How are you attributing causality etc.).

line 420-421: If I understand this paragraph correctly, the paper finds similar strategies (e.g., uncertainty guided refinement) appear in solutions to multiple problems. The paper then concludes "These recurring strategies suggest that the deep research agent not only extracts task-specific insights but also steers the coding agent toward generalizable algorithmic principles.". But how do we know these recurring patterns would not have appeared without the deep research? For example, If I ask chatGPT to find minima of mathematical functions, then the minimization algorithms will also often be similar ones

4.4: This partially answers my point about table 2. This seems to be a much more fair comparison. However, based on the two datapoints of best score, I would conclude that the improvement of adding deep research is on the order of 2-9%.
If the central claim of the paper is that adding deep research to coding agents leads to great improvement, this table should list all problems (rather than just 2) and these should be the improvements quoted. You also write "Algorithm evolution based solely on LLM internal knowledge shows limited progress (...) or yield marginal gains despite deeper evolution (Circle Packing). " But isn't the score improvement without deep research 2.735 vs the one with deep research 2.981, both improved from baseline of 0.389? This doesn't sound like marginal improvement to me. In my opinion, this table/ablation study should be the main result of the paper if

---

### Official Review · Reviewer_Rkic · 2025-11-09

**Soundness:** 3
**Presentation:** 3
**Contribution:** 1
**Rating:** 2
**Confidence:** 4

**Summary:**

This paper proposes deepevolve, which is a procedural including the fundamental idea of alphaevolve, and also incorporating deep research. The procedure consists of writing -> coding -> evaluation -> evolutionary database -> planning -> searching. The evolutionary database reflects more of the alphaevlove part, while the remaining part incorporates deep research.

**Strengths:**

1. This paper conducts experiments on many topics, such as chemistry, mathematics, biology.
2. The presentation is generally good, with clear figures.

**Weaknesses:**

It is unclear on what is the novelty of this paper. The procedure is not novel, but widely used in scientific discovery research that incorporates experimental feedback, such as alphaevolve. The main idea is about to connect the idea of alphaevolve and deep research. Here deep research can be seen as a more powerful version of an LLM itself that can connect to online search and more reasoning steps. So fundamentally the idea is to adopt more powerful LLMs for alphaevolve. In fact, for any LLM-involved workflow, it is nearly guarantteed to receive better performance if we just replace the LLM included with deep research. In this sense, I'm not sure what new knowledge I can learn from this paper.

**Questions:**

What can we learn from this paper? What new knoweldge it introduces to the field?

---

### Meta-Review · Area_Chair_wcDS · 2025-12-08

**Summary:**

Thank you to the reviewers for your valuable suggestions from multiple perspectives. Overall, I think their main problems at present lie in:

- Lack of innovation.
- Inadequate experimental results.
- Some descriptions are unclear.
- Some recent work is not discussed.
- The efficiency of the method is not clarified.

**Reviewer Concerns:**

The author did not respond to the reviewer's questions. Considering the comments of the four reviewers, I think the paper is clearly below the acceptance threshold.

**Reviewer Scores:**

All reviewers will keep the original score.

---

### Decision · Program_Chairs · 2026-01-26

Reject